# Genomic Landscape of Hodgkin Lymphoma

**DOI:** 10.3390/cancers13040682

**Published:** 2021-02-08

**Authors:** Magdalena M. Brune, Darius Juskevicius, Jasmin Haslbauer, Stefan Dirnhofer, Alexandar Tzankov

**Affiliations:** Institute of Medical Genetics and Pathology, University Hospital Basel, Schönbeinstrasse 40, CH 4031 Basel, Switzerland; magdalena.brune@usb.ch (M.M.B.); darius.juskevicius@usb.ch (D.J.); jasmindionne.haslbauer@usb.ch (J.H.); stefan.dirnhofer@usb.ch (S.D.)

**Keywords:** Hodgkin lymphoma, cell lines, cell enrichment, 9p24 locus, JAK/STAT, NF-κB, PDL1, PDL2, PI3K/AKT/mTOR, Epstein-Barr virus, mutations, genomes, epigenetics

## Abstract

**Simple Summary:**

Hodgkin lymphoma (HL) is composed of many reactive and only a few cancer cells, so-called Hodgkin and Reed-Sternberg (HRS) or lymphocyte predominant (LP) cells. Due to the scarcity of these cells, it was difficult to perform high-throughput molecular investigations on them for a long time. With the help of recently developed methods, it is now possible to analyze their genomes. This review summarizes the genetic alterations found in HRS and LP cells that impact immune evasion, proliferation and circumvention of programmed cell death in HL. Understanding these underlying molecular mechanisms is essential, as they may be of prognostic and predictive value and help to improve the therapy especially for patients with recurrent or treatment-resistant disease.

**Abstract:**

Background: Hodgkin lymphoma (HL) is predominantly composed of reactive, non-neoplastic cells surrounding scarcely distributed tumor cells, that is, so-called Hodgkin and Reed-Sternberg (HRS) or lymphocyte predominant (LP) cells. This scarcity impeded the analysis of the tumor cell genomes for a long time, but recently developed methods (especially laser capture microdissection, flow cytometry/fluorescence-activated cell sorting) facilitated molecular investigation, elucidating the pathophysiological principles of “Hodgkin lymphomagenesis”. Methods: We reviewed the relevant literature of the last three decades focusing on the genomic landscape of classic and nodular lymphocyte predominant HL (NLPHL) and summarized molecular cornerstones. Results: Firstly, the malignant cells of HL evade the immune system by altered expression of *PDL1/2*, *B2M* and MHC class I and II due to various genetic alterations. Secondly, tumor growth is promoted by permanently activated JAK/STAT signaling due to pervasive mutations of multiple genes involved in the pathway. Thirdly, apoptosis of neoplastic cells is prevented by alterations of NF-κB compounds and the PI3K/AKT/mTOR axis. Additionally, Epstein-Barr virus infection can simultaneously activate JAK/STAT and NF-κB, similarly leading to enhanced survival and evasion of apoptosis. Finally, epigenetic phenomena such as promoter hypermethylation lead to the downregulation of B-lineage-specific, tumor-suppressor and immune regulation genes. Conclusion: The blueprint of HL genomics has been laid, paving the way for future investigations into its complex pathophysiology.

## 1. Introduction: Searching and Finding the Needle in the Haystack

Hodgkin lymphoma (HL) is unique in many aspects. In contrast to many other malignant tumors, HL not only escapes immunologic control but uses and recruits the immune system for its purposes, which leads to a significant and heterogeneous tumor microenvironment. Indeed, malignant lymphocyte predominant (LP) cells in nodular lymphocyte predominant HL (NLPHL), as well as Hodgkin and Reed-Sternberg (HRS) cells in classic HL (cHL), represent only a minor component (typically less than 1%) of the affected tissues. Investigation of these scarce neoplastic cells, therefore, resembles the search of the needle in the haystack, substantially impeding the analysis of the respective tumor cell genomes, as high-throughput molecular research methods require high tumor cell purity and substantial input DNA content. However, recent developments in the field may facilitate the study of HL genomics.

Much insight into the genetics of HL has been gained through the investigation of cell lines, making the application of, for example, whole-exome sequencing possible [1]. However, the validity of these investigations is limited as they are not representative of the complex intratumoral heterogeneity in HL. Particularly in cHL, almost all cell lines derive from refractory and relapsing cases, by no means reflecting the great majority of clinical courses [2]. Furthermore, there is only one existing NLPHL cell line (DEV) and some bona fide HL cell lines that have been misidentified as such [3,4]. In contrast, laser capture microdissection (LCM) of neoplastic LP and HRS cells from affected tissues represents a more realistic approach and, until very recently, has been the most widely used method in genetic studies of cHL. With the help of this technique, recurrent genomic gains and losses, as well as mutations of single candidate genes, have been identified (e.g., [5,6]). Nevertheless, LCM has important disadvantages that limit its use. Apart from being highly labor-intensive, this method involves high-energy laser radiation, potentially leading to DNA damage. Furthermore, tumor cells are collected out of tissue slides, which contain truncated nuclei possibly leading to impaired assay quality [7]. An alternative method using flow cytometry, enabling rapid isolation and high purification of HRS cells out of fresh or frozen tissue samples was introduced by Fromm et al. [8]. Based on this protocol, the first whole-exome sequencing study of HRS cells was performed in 2015 [9]. In 2018, our group developed a novel fluorescence-activated cell sorting (FACS)-based enrichment technique for the isolation of HRS cells out of formalin-fixed and paraffin-embedded (FFPE) archival tissues [7]. This involves DNA enrichment followed by whole-genome amplification and the application of a customized lymphoma panel assay of 68 genes [10]. Although the enrichment of tens of thousands of tumor cells is possible without potential laser radiation-induced DNA damage, the preservation of sufficient tissue antigenicity and cytometric recognition of specific signals remain obstacles in FFPE [7,11]. Another promising and elegant approach is the analysis of circulating tumor DNA (ctDNA) of patients with cHL as reported by Spina et al. [12] and most recently by Desch et al. for pediatric HL patients [13]. This capture-based technology can noninvasively detect mutations in HRS cells, tracking their clonal evolution in the course of therapy as well as monitoring minimal residual disease.

With the help of these methods, a general framework of the cellular pathways and genetic alterations of HL have been elucidated in the last three decades. In 1994, the B-cell lineage origin of HRS cells was identified by demonstrating clonally rearranged immunoglobulin heavy- and light-chain (*IG*) genes bearing crippling mutations [14]. Under physiological circumstances, B-cells bearing such mutations would rapidly be eliminated by apoptosis. Thus, in order to undermine apoptosis-inducing mechanisms, HRS cells manipulate key cellular signaling pathways (mainly JAK/STAT and NF-κB), introducing a multitude of salvage pathways (immune evasion and infection/reactivation of Epstein Barr virus (EBV)) as well as promoter hypermethylation of genes (e.g., [15,16,17,18,19]). Thus, multiple-aptly called pervasive mutations have been identified in HL [11,16]. In the next sections, known genomic aberrations in HL will be discussed in the light of affected pathways (Figure 1).

## 2. Immune Evasion

Several genes involved in the development of hematolymphoid malignancies are located at the 9p24 locus. This includes key targets of immune checkpoint inhibition such as programmed death ligands 1 and 2 (*PDL1/PDL2*), which evokes potential therapeutic interest (rev. in [20] and in other contributions within this special issue). Most investigated cases of cHL show genetic alterations of *PDL1/2*, most commonly copy number gains and amplifications (up to 55% and 35%, respectively) [15]. In cHL—mainly the nodular sclerosis subtype—these copy number gains were found to correlate with higher expression of *PDL1* as determined by immunohistochemistry (Figure 2A). They represent the hallmark of tumor-induced immune modulation mainly impeding effector T-cell proliferation and activation as well as stimulating immunosuppressive regulatory T-cells [15,20,21,22]. In a minority of cases, structural abnormalities of 9p24 have been found in cHL (Figure 2B, insert), leading to translocations of *PDL1/2* to several partners [23]. Single nucleotide mutations, insertions and deletions of *PDL1/2* are not (yet) found to play a major role in lymphomagenesis [20].

Importantly, 9p24 alterations, especially copy number gains, were associated with inferior outcome in conventionally treated patients [15] but were an indicator for response and superior progression-free survival—to PD1/PDL1 immune checkpoint inhibition-based immunotherapy [24]. Along with very convincing results of a prospective trial with nivolumab in relapsed or refractory cHL [25], this was founding for treatment advances in such instances as nicely reviewed [26], and may lead to first-line therapy paradigm changes in cHL. Furthermore, a very recent work highlighted the importance of a broad baseline T-cell repertoire for successful immune-checkpoint inhibitor treatment, being most effective in patients with therapy-associated diversity increase in the CD4^+^ compartment and in those with an abundance of activated natural killer cells and a newly identified CD3^−^CD68^+^CD4^+^GrB^+^ subset of innate immune cells, which may function as direct cytotoxic effectors in even the absence of major histocompatibility complex (MHC) class I (the latter being characteristic of cHL; see below) [27].

Inactivating mutations of the beta 2 microglobulin gene (*B2M*) also play an instrumental role in immune evasion, influencing the assembly of MHC class I and thus altering tumor cell “visibility” for effector cells [1]. Indeed, *B2M* is the most commonly mutated or deleted gene in up to 70% of studied cHL cases [9,11]. Furthermore, its deficiency is associated with the nodular sclerosis subtype, pointing towards its potential influence on the tumor microenvironment [7,9,16]. Wienand et al. additionally detected mutations or deletions of *HLA-B* in approximately 15% of cHL, representing another potential mechanism of MHC class I assembly dysregulation. Moreover, the *MHCI* (and *MHCII*) loci at 6p21 are among the commonly deleted in cHL [5,7,11]. Interestingly, a decrease of MHC class I expression is associated with inferior clinical outcome after standard chemotherapy, but not immune-checkpoint inhibition. To be comprehensive, Epstein-Barr virus (EBV) positive cHL have significantly higher MHC class I expression on HRS cells than EBV-negative cases [11,28]. In contrast to MHC class I, the expression of MHC class II is predictive for response to PD1-blockade in cHL [24], fitting well with the above-mentioned observations on the central role of immune responses linked to the CD4^+^ cellular compartment.

Finally, the MHC class II transactivator *CIITA* has been identified to be involved in a gene fusion in cHL cell lines and in 15% of investigated clinical cases [17]. Genomic aberrations in *CIITA* result in a downregulation of surface MHC class II expression as well as overexpression of PDL1/PDL2, hampering anti-tumor immune response.

A complex network of cytokines and chemokines secreted by both malignant and reactive cells orchestrates the interaction between HRS and LP cells, respectively, and the surrounding microenvironment [29]. One component of this network is the immunosuppressive effect of transforming growth factor-beta (TGF-β) on tumor-infiltrating lymphocytes. Until recently, it was unclear why HRS and LP cells remain unaffected by the anti-neoplastic properties of TGF-β. Previous studies on diffuse large B-cell lymphoma (DLBCL) revealed SMAD1 as a key messenger in the tumor-suppressive signaling axis of TGF-β [30]. In concordance with this study, our group was able to show a lack of SMAD1 expression due to hypermethylation of its promoter region in LP and HRS cells of almost all studied clinical cases (14/14 NLPHL cases, 100% and 138/143 cHL cases, 97%). Most interestingly, this mechanism was reversible in an affected cell line by treatment with decitabine, a DNA methyltransferase inhibitor [19].

## 3. Pervasive JAK/STAT Signaling

Another gene located at 9p24 is Janus kinase 2 (*JAK2*), which has gained much attention in the context of myeloproliferative diseases. As *JAK2* is localized in close proximity to *PDL1/2*, it is most often encompassed in amplifications of the 9p24 region [31,32]. JAK2 belongs to a family of cytoplasmic tyrosine kinases that activate its target genes via phosphorylation of signal transducer and activator of transcription (STAT) factors. Furthermore, JAKs are capable of direct chromatin remodeling by histone phosphorylation and are able to activate other intracellular pathways (e.g., AKT) [33]. Although activating point mutations of *JAK2*, which are common in myeloid neoplasms, are very rarely found in lymphomas, the enhanced activity of the JAK/STAT signaling pathway by 9p24 amplification seems to play an important role in the development of cHL as our own data and other studies suggest [16,31]. The increased gene dosage of *JAK2* indeed seems to have a functional effect, as cases with 9p24 amplification show higher amounts of phosphorylated—and thus activated—JAK2 (Figure 2B) and STAT3 [31]. The central importance of the JAK/STAT cascade to HL is reflected by the presence of various alterations of multiple genes encoding multiple members of the pathway. Inactivating mutations, deletions and—especially in pediatric cHL [13]—inactivating translocations of *SOCS1* and, rarely, *SOCS6*, both negative regulators of JAK/STAT signaling, can be found in up to 50% of NLPHL [34] and 60% of cHL [7,11,16,35]. This consecutively leads to the accumulation of various nuclear STATs in HL (Figure 2D–F). Bona fide activating mutations of the nuclear shuttle protein *XPO1* also contribute to the nuclear accumulation of STATs and have been found in 18%, 24% and 26% of the studied cHL cases, respectively [11,16,36], and patients with the detectable hotspot mutation *XPO1* E571K even tended towards a shorter progression-free survival [36]. XPO1 (or CRM1) is responsible for the nucleo-cytoplasmatic transport of more than 200 proteins, among them p53 and phosphorylated STATs [37,38]. Abnormal expression of this now therapeutically targetable protein has been found to worsen the prognosis of DLBCL patients that already display prognostically unfavorable factors such as BCL2 overexpression but also predicts a better response to the clinically available XPO1 inhibitor selinexor in this particular subgroup [39]. Importantly, Tiacci et al. treated cHL cell lines with selinexor, which resulted in growth inhibition and induction of apoptosis in cells with an E571K hotspot mutation [16].

Another commonly affected target is *STAT6* (Figure 2C); gains of its coding chromosomal region 12q13 and activating missense mutations are found in approximately 50% and 40% of cHL, respectively [5,7,12,16]. Importantly, even cooperative mutations in *SOCS1* and *STATs* have been found in cHL [16].

Activating mutations in the cytokine receptor *CSF2RB* were found to enhance JAK/STAT signaling in up to 20% of cHL [11]. Other identified genetic alterations impair JAK2 dephosphorylation; these are abrogating mutations and deletions of *PTPN1* and promoter hypermethylation of *PTPN6* (*SHP1*) in subsets of cHL [40,41]. Additionally, rare fusions of the *JAK2* locus that may lead to constitutively active JAK2 have been described in cHL (e.g., *JAK2-SEC31*) [31,42].

Most importantly, there seems to be an additive effect of *PDL1/2* alterations and the activation of the JAK/STAT signaling pathway: Green et al. demonstrated that JAK2/STAT synergistically enhance PDL1 expression, thus further fueling a vicious circle [22].

The lysine (K)-specific demethylase 4C (*KDM4C*) gene, also known as *JMJDC2*, is similarly found at the 9p24 locus. The amplification of this chromatin remodeler has been suggested, in addition, and in synergy to JAK2, to have a pro-oncogenic effect on HL, among others upregulating MYC expression in lymphoma cells [32,43].

## 4. Disruption of the NF-κB Pathway

The binding of the transcription factor NF-κB enhances cell proliferation and survival and diminishes the effect of proapoptotic signals. Aberrant NF-κB pathway compounds are found in about 50% of cHL cases, consistent with the prominent role of NF-κB signaling in HRS cells. This includes inactivating mutations and deletions of *TNFAIP3* coding for A20, one of the key inhibitors of the canonical NF-κB pathway, in up to 44% of analyzed cell lines and cHL cases, especially of the nodular sclerosis subtype and in pediatric cases [9,12,13,16,44]. Additionally, other rare activating mutations of *IKBKB* and inactivating mutations (rarely deletions) of the inhibitors *NFKBIE* and *NFKBIA* have been identified [11,12,13,16,45,46,47,48].

Our own studies [7] and previous findings [9,49,50] revealed gains of the entire short arm of chromosome 2 (2p) in approximately 60% of investigated cHL samples. Amongst other genes, this chromosomal region contains *REL*, a subunit of the canonical NF-κB pathway. In another 30% of the cases, *CARD11*, another upstream component of the NF-κB pathway, has been found to be mutated in cHL [7].

## 5. Alterations in PI3K/AKT/mTOR Pathway and Impairment of Cytokinesis

The PI3K/AKT/mTOR pathway is one of the key cell cycle regulators. It is an important target for alterations in cancer cells leading to a net effect of uncontrolled proliferation and decreased apoptosis [51]. Perturbations of this signaling cascade have been found in approximately 45% of cHL [12]. For instance, *GNA13,* which encodes the G13 protein alpha subunit, a tumor suppressor that inhibits AKT phosphorylation and helps to control proliferating germinal center B cells [16] is a recurrent target (25%) of mutations in cHL [11,13,16]. Inactivating mutations encompass missense, nonsense, as well as frameshift mutations, and are tightly associated with *STAT6* alterations. Inactivation of GNA13 results in decreased apoptosis of dysfunctional germinal center B-cells and their spread outside the lymphoid follicle [52]. Aberrations of the PI3K/AKT signaling activity may be linked to inactivating *ITPKB* mutations, found in slightly over 25% of cHL [12]. Normally, ITPKB dampens PI3K/AKT signaling by upregulating IP4, a soluble pathway antagonist [53].

Importantly, Hodgkin cells generate multinuclear Reed-Sternberg cells—a cytomorphological hallmark of cHL—due to incomplete cytokinesis [54,55]. Indeed, mutations of *GNA13* also result in dysfunctional microtubule dynamics during cytokinesis by interacting with RhoA [56,57]. In addition, during cytokinesis, the DNA-binding protein CDH1 is excluded from the nucleus and is reincorporated during telophase [58]. The identified posttranslational alterations of *CDH1* in cHL may impair this process, leading to impaired cytokinesis [41]. Furthermore, *DNAH12*, a gene encoding for dynein axonemal heavy chain 12 essential for microtubule motor activity, is frequently mutated in cHL [11].

## 6. Other Genetic Aberrations

*TP53* mutations are found in almost 10% of cHL cases and are associated with a higher number of overall mutations, stressing TP53′s role as the “guardian of the genome” [7,16]. One group even identified *TP53* to be the most commonly mutated gene in their investigated cHL series with an incidence of slightly over 20% [59]. Interestingly, a loss of RMB38, an RNA-binding protein that restrains the translation of wild-type p53 mRNA, has been shown to drive lymphomagenesis in murine models via increased mutant TP53 expression and decreased expression of PTEN; truncating and missense mutations of *RBM38* have been found in approximately 15% of investigated cHL [11,60].

Structural variants of *ETV6*, a commonly rearranged locus in hematological and solid malignancies, were found in just over 15% of cHL cases [11]. Although the exact oncogenic mechanism by which *ETV6* translocations contribute to lymphomagenesis is still to be determined, it is assumed to lead to a loss of ETV6-mediated transcriptional control mechanisms [61].

Another pathway recurrently altered by genomic lesions in HRS cells is NOTCH signaling, found to be cumulatively mutated in up to 20% of the investigated cases including alterations in *SPEN* (12.5%), *NOTCH1/2* (2.5%, respectively) and *FBXW7* (7.5%) [12].

## 7. Epstein-Barr Virus

Approximately 40% of cHL in Western nations are associated with latent EBV infection and presumed to be reliant on its latent membrane antigen 1 (LMP1) [62,63]. LMP1 can activate NF-κB and PI3K/AKT/mTOR signaling by simulating an active CD40 receptor [64,65]. In addition, PDL1 expression can be induced by LMP1 via the activation of STATs, particularly STAT3, introducing an alternative pathway of immune evasion [66]. In general, adult EBV-positive cHL seems to be less dependent on genetically altered intracellular pathways, showing significantly lower numbers of somatic mutations [11,16], which is not that pronounced in pediatric instances [13]. This is analogous to what has been shown in other EBV-associated B-cell lymphomas, in which EBV infection may “substitute” for mutations of genes encoding for immune escape mechanisms [67,68]. Importantly, when corrected for the EBV-status, no correlation was found between histological subtype, mutational burden and mutational types in cHL per se [11,16].

## 8. Epigenetics in cHL

Promoter hypermethylation has previously been identified as the underlying mechanism by which HRS cells downregulate B-lineage–specific genes (Figure 3A) such as *PU1*, *BOB1*, *SYK*, *CD19* and *CD79B* [69]. Interestingly, data seems to suggest a pathophysiological link between epigenetics and EBV: the frequency of promoter hypermethylation was—similar to what has been shown for mutations—higher in EBV-negative compared to EBV-positive cases, but could occur in both [41] and, in the latter, has been suggested to be dependent on LMP1 through the activation of DNA-methyltransferases [70]. In addition to *SMAD1* (see above) (Figure 3B), promoter hypermethylation of seven cancer-related genes has been found in cHL: *CDKN2A* (77%), *RASSF1A* (59%), *CDH1* (51%), *DAPK* (45%), *GSTP1* (43%), *SHP1* (also called *PTPN6*) (38%) and *MGMT* (24%) [41]. These genes are involved in the induction of apoptosis, cytokinesis, chromatin and DNA or RNA functions, which are all known to be perturbed in cHL. Finally, Spina et al. found epigenetic changes, mostly related to mutations of *ARID1A* and *KMT2D*, cumulatively affecting 35% of patients [12].

## 9. NLPHL

Little is known about the genetic landscape of LP cells in NLPHL. They were found to be very closely related to the neoplastic cells in T-cell/histiocyte-rich large B-cell lymphoma, which is clinically a completely different entity [71]. On the other hand, comparisons of the gene expression patterns of LP and HRS cells revealed surprising similarities, as both show constitutive activation of NF-κB and JAK/STAT signaling (mainly due to mutations of *SOCS1*) [34,72]. Yet, distinct mutations frequently observed in cHL, such as *TNFAIP3* and *NFKBIA*, seem to be less frequent in NLPHL, while others, such as *SGK1*, *DUSP2* and *JUNB*, appear to be more common. Genes targeted by somatic hypermutation (a process that is ongoing in NLPHL but to a lesser extent in cHL), such as *PAX5*, *PIM1*, *RHOH* and *MYC*, are more frequently altered in NLPHL compared to cHL [73,74,75]. Recently, a recurrent deletion in the chromosomal region 9p11 and translocations affecting *BCL6* in a subset of NLPHL have been identified [72,76,77]. Both of these aberrations have not been found in cHL.

## 10. Conclusions

In summary, the pathogenesis of HL relies on several important pathophysiological principles. Firstly, immune escape is initiated by altered expression of PDL1/2, B2M, MHC class I and MHC class II. Undermining PDL1/2 signaling and consequently overcoming this immune evasion has become the focus of PD1/PDL1 driven immune checkpoint inhibition studies, as was successfully shown in initial clinical trials for relapsed or refractory HL [25]. Although the prognostic and predictive significance of PDL1 status in HL is not yet fully understood, there is evidence for shorter progression-free survival and an increased incidence of advanced stage disease in patients with 9p24 amplifications [15]. Secondly, constitutively active JAK/STAT signaling as a result of an array of mutations, such as *STAT6* and *SOCS1*, plays a central role in cHL carcinogenesis. Thirdly, members encoding for NF-κB compounds (and PI3K/AKT/mTOR) are subjected to mutations in HL; these targets typically promote growth and proliferation under physiological conditions but also prevent apoptosis that would eliminate B-cells with crippling *IG* mutations as observed in HRS cells. Finally, JAK/STAT and NF-κB can be simultaneously activated by LMP1 in EBV-positive cases, eventually leading to PDL1 overexpression, consequently leading to the above-mentioned scenarios.

With the advancement of robust molecular methods enabling investigations of neoplasms with tumor cell paucity, our understanding of the genomic landscape of HL has greatly expanded (Figure 1). The significance of this gained knowledge concerning prognostic and predictive value needs further elucidation. Future investigations should feature larger FFPE cohorts with a particular focus on refractory and relapsed HL.

## Figures and Tables

**Figure 1 cancers-13-00682-f001:**
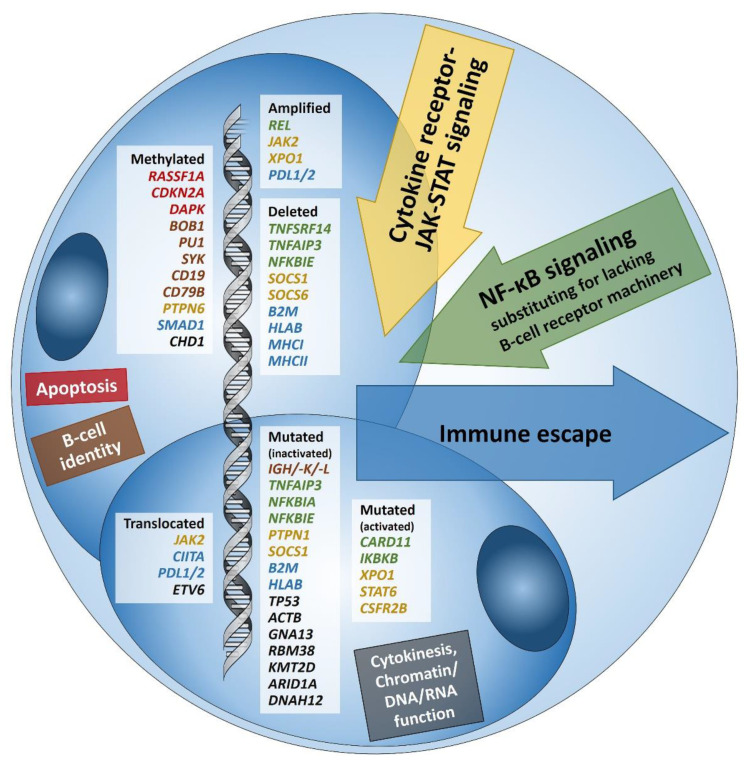
Summary of known genetic aberrations in classic Hodgkin lymphoma (cHL) arranged according to aberration type and color-coded according to the affected cellular process that they dysregulate in Hodgkin and Reed-Sternberg cells; genes encoding for proteins related to apoptosis are in red, to B-cell identity—in brown, to cytokine (mainly JAK-STAT) signaling—in orange, to NF-κB signaling—in green, to immune escape—in blue, and to cytokinesis, chromatin/DNA/RNA functions—in black; inactivating translocations of *SOCS1* that are characteristic of pediatric cHL [13] are not sown.

**Figure 2 cancers-13-00682-f002:**
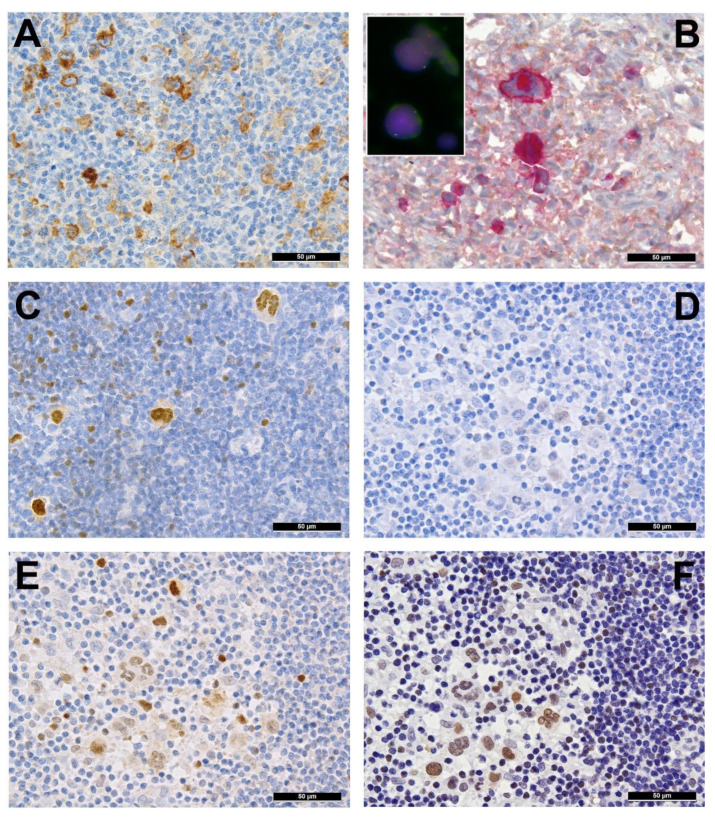
(**A**) PDL1 overexpressing Hodgkin and Reed-Sternberg (HRS) cells in a case of *PDL1/2* amplified classic Hodgkin lymphoma (cHL). (**B**) HRS cells expressing phosphorylated (p) JAK2 in a *JAK2* rearranged cHL (insert with split red and green FISH signals corresponding to the rearranged allele and one fused yellow signal corresponding to the wild type allele of the *JAK2* gene in the respective large HRS cell-equivalents utilizing a break-apart *JAK2* probe). (**C**) pSTAT6 overexpressing HRS cells in a case of *STAT6*-mutated cHL. (**D**) Expression of pSTAT6 only in a few HRS cells as compared to. (**E**) pSTAT5 and, particularly, pSTAT3. (**F**) in a case of *SOCS1*-mutated cHL.

**Figure 3 cancers-13-00682-f003:**
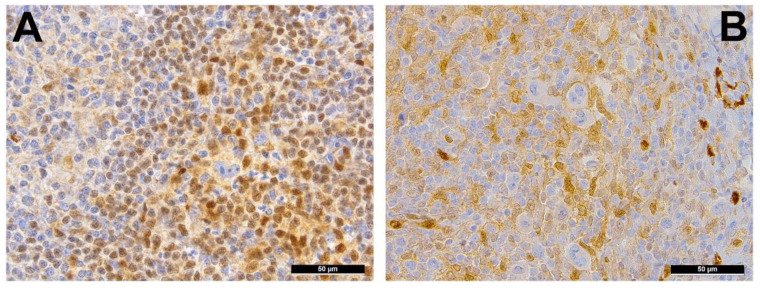
(**A**) BOB1-negative Reed-Sternberg (RS) cell surrounded by BOB1 positive B-cells; *BOB1* inactivation due to promoter hypermethylation of the respective gene in classic Hodgkin lymphoma (cHL). (**B**) SMAD1 negative RS cells surrounded by SMAD1 positive lymphocytes; *SMAD1* is known to be hypermethylated in cHL and nodular lymphocyte-predominant Hodgkin lymphoma.

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
