# Peer review of "Genomic Landscape of Hodgkin Lymphoma"

_cancers, 2021, doi:10.3390/cancers13040682_

Round 1

Reviewer 1 Report

This is an excellent, well-organized comprehensive review of what has been learned about the biology of both classical Hodgkin lymphoma and nodular lymphocyte predominant Hodgkin lymphoma with new methods. As a clinician, it will be an invaluable reference.

Author Response

We thank to ther referee for her/his time spent with our manuscript and for her/his kind comment.

Reviewer 2 Report

This is an interesting review on immune checkpoint inhibition in classical Hodgkin lymphoma (HL), but the manuscript may be improved is several ways:

1-The manuscript should be more focused on HL. As an example, experience with CTLA-4 inhibitors may be restricted to a short paragraph at the end given the very limited data in HL. The same option could be also taken for the paragraphs on immune checkpoints inhibitors and Blockade of the PD1/PD-L Axis.

2-It could be important to report the data on nivolumab and pembrolizumab in parallel and with the same interest since the 2 drugs have the same mechanisms of action and are both FDA and EMA approved in this indication.

Retrospective studies (like the one from Herbaux (75)) should be presented with a warning on the robustness of the data.

3-first line therapy in HL is really difficult for new drug development. This point could be really interesting to discuss (way to improve therapy, why bleomycin is skipped….).

4-combination therapies tested in HL  could be discussed and their design scientifically presented (rational, ongoing studies..)

Author Response

We thank to the referee for her/his time spent with our manuscript and for her/his valuable suggestions.

Since the special issue of Cancers for which our review has been invited will contain at least two papers with a special focus on recent advances in therapy of Hodgkin lymphoma (Pathogenetic markers in Hodgkin lymphoma evolving in future theragnostic targetsï¼›Innovative therapies in childhood Hodgkin Lymphoma) and due to space limitations as well as in the face of a very nice and comprehensive review paper un Cancers addressing recent respective clinical trials (Voorhees, T.J.; Beaven, A.W. Therapeutic updates for relapsed and refractory classical Hodgkin lymphoma. Cancers (Basel) 2020, 12, 2887. doi: 10.3390/cancers12102887.), we only shortly addressed all 4 points raised by the referee - pages 3-6, lines 105-129.

Reviewer 3 Report

The review by Brune and colleagues provides an up-to-date overview about the genetic lesions identified in the tumor cells of classical and lymphocyte predominant Hodgkin lymphoma. As the knowledge about the genetic lesions was recently considerably extended through the first whole exome and targeted deep sequencing studies, this is a timely topic.

Main criticism:

In many instances, three recent exome sequencing studies are cited as reference for the detection of mutations in HRS or LP cells. However, in numerous instances, the genes were already identified as mutated in Hodgkin lymphoma in earlier candidate gene approaches in isolated HRS or LP cells. Appropriate credit should be given for these earlier studies by citing them as well. This hold true in particular for the genes SOCS1, XPO1, TNFAIP3, NFKBIA, NFKBIE, REL (gains), JAK3 (gains), BCL6 translocations (LP cells). Furthermore, for STAT6, also gains have been identified in HRS cells (Hartmann et al.,).

It is recommended to also mention the recent genetic study of pediatric Hodgkin lymphoma patients (A.-K. Desch et al., 2020, Leukemia).

Author Response

We thank to the referee for her/his time spent with our manuscript and for her/his valuable suggestions.

She or he is absolutely right to request citations of earlier studies that contributed to the identification of mutations in Hodgkin lymphoma. We therefore added additional references for the suggested genes; new references 32, 33, 41-47, 73 and 74.

We also implemented the study on pediatric instances; new reference 13.

Reviewer 4 Report

The authors summarize in this review the genetic alterations of Classical and Lymphocyte predominant Hodgkin Lymphoma and their impact on immune evasion and on survival  of the neoplastic clone.

The review is well written, concise but clear and exhaustive both in the alterations described and in their role in the biopathology of the disease. The text is accompanied by appropriate histological diagrams and images. the bibliography is complete complete.

Please note the following:
Inset figure 2: the FISH picture is poorly defined.

Author Response

We thank to the referee for her/his time spent with our manuscript and for her/his kind feedback.

As suggested, we added explanatory notes to the FISH insert in figure 2.

Round 2

Reviewer 2 Report

this is an interesting review on genomic abnormalities in HL. This manuscript nicely prsent that HL in a disease with many oncogenic abnormalities leading to abnormal immune response against HL cells.

It will be important to present data on IO drugs response (Nivolumab or pembrolizumab) and these abnormalities (PDL1 expression, EBV presence..). The authors shiould discuss a model on th potential relation of IO drugs and these abnormalities (as an example MHC genetic alteration is a mechanism of anti-PD1 resistance in melanoma, what about HL?)

Some data of immune exclusion exist in HL using multiplex technology on tissu samples, it could also be interesting to discuss this observation in the light of genetic datas?

Author Response

We are grateful to referee 2 for her/his encouraging notice that this is an interesting review on genomic abnormalities in HL. 

In the second revised version, we addressed the issues linked to the predictive value of these genomic genomic abnormalities on IO drugs responses as well as on the role of MHC genetic alterations as mechanism of anti-PD1 sensitivity (page 5, lines 125-152 of the revision) as required by the reviewer. We would like to particularly thank for her/his advice in this consideration, since we feel that addition of this information increased the comprehensiveness of the paper.

Round 3

Reviewer 2 Report

no comment